# A Randomised Controlled Trial of a Caregiver-Facilitated Problem-Solving Based Self-Learning Program for Family Carers of People with Early Psychosis

**DOI:** 10.3390/ijerph17249343

**Published:** 2020-12-14

**Authors:** Wai Tong Chien, Daniel Bressington, Dan I. Lubman, Thanos Karatzias

**Affiliations:** 1The Nethersole School of Nursing, Faculty of Medicine, The Chinese University of Hong Kong, Hong Kong, China; 2College of Nursing and Midwifery, Charles Darwin University, 1 Ellengowan Drive, Casuarina NT810, Australia; daniel.bressington@cdu.edu.au; 3Addiction and Behavioral Research, Eastern Health School, Monash University, Victoria 3800, Australia; dilubman@gmail.com; 4School of Health and Social Care, Edinburgh Napier University, Edinburgh EH11 4BN, UK; T.Karatzias@napier.ac.uk

**Keywords:** problem solving, psychoeducation, randomised controlled trial, psychosis, self-learning

## Abstract

Facilitated self-help and problem-solving strategies can empower and support family carers to cope with caregiving for people with severe mental illnesses. This single-blind multi-site randomised controlled trial examined the effects of a five-month family-facilitated problem-solving based self-learning program (PBSP in addition to usual care), versus a family psychoeducation group program and usual psychiatric care only in recent-onset psychosis, with a six-month follow-up. In each of three study sites (integrated community centres for mental wellness), 114 people with early psychosis (≤5 years illness onset) and their family carers were randomly selected and allocated to one of three study groups (*n* = 38). Caregiving burden (primary outcome) and patients’ and carers’ health conditions were assessed at recruitment, and one-month and six-months post-intervention. Overall, 106 (94.7%) participants completed the assigned intervention and ≥1 post-test. Generalised estimating equations and subsequent contrast tests indicated that the PBSP participants showed significantly greater improvements in carers’ burden, caregiving experiences and problem-solving ability, and patients’ psychotic symptoms, recovery, and duration of re-hospitalisations over the six-month follow-up, compared with the other two groups (moderate to large effect size, η^2^ = 0.12–0.24). Family-assisted problem-solving based self-learning programs were found to be effective to improve both psychotic patients’ and their carers’ psychosocial health over a medium term, thus reducing patients’ risk of relapse.

## 1. Introduction

Psychosis is a major disabling and disruptive mental illness, accounting for over 30% of psychiatric patient populations in Hong Kong and globally [1,2], and often having a high risk of relapse in the first five years of illness [3]. As recommended by the National Institute for Health and Clinical Excellence, U.K. [4], and consistent with the findings of systematic reviews [5,6], effective intervention for early-stage psychosis should include effective medication management, psychoeducation, active rehabilitation, and family support services to provide adequate informational and psychosocial support for both patients and their family carers.

Family carers living with a person with psychosis frequently face high levels of burden and family conflict, thus adversely affecting their caregiving and general well-being. In particular, family carers with a high caregiving burden in an absence of effective coping strategies and resources can manifest high expressed emotions towards the patient [7,8], and in turn increase the patient’s risk of relapse. Therefore, family-oriented interventions with supportive and educational elements can be important to meet carers’ health needs and caring roles, particularly being first-time caregivers to their young relative with psychosis [6,9,10].

Systematic reviews of controlled trials of family interventions in early psychosis concluded that psychosocial interventions such as cognitive-behavioural therapy and psychoeducation programs could significantly improve families’ psychosocial functioning and knowledge of the illness/treatment [11,12,13], but only result in small or non-significant effects on other family outcomes such as caregiving burden and positive/gaining experiences, and over a longer-term follow-up. Despite having high levels of psychological and resource supports by health professionals, some families have reported difficulties in regularly participating in psychoeducation sessions because of time constraints, a sense of social/internal stigmatisation attached to mental illness, and negative experiences in help-seeking [7,14]. With scarce family support services, family caregivers need to improve personalised illness self-management strategies and develop partnerships when coordinating care for their relative with psychosis [15].

There has been increasing research/service needs for self-help programs in empowering patients with chronic mental and physical illnesses such as depression, anxiety and eating disorders [16,17], and their family carers [10,18]. Replacing professional input with self-management may enhance intervention feasibility and/or accessibility and reduce its delivery costs. However, only few family interventions in early-stage psychosis have focused/emphasised self-help and problem-solving skill acquisitions; most controlled trials of the interventions reported high attrition rates (30–60%) [5,6,11]. Overcoming the main barriers to family interventions such as stigmatisation, over-dependence, and low flexibility/acceptability to services [8,19], problem-solving based self-help manual-reading and daily practices can be a structured user-friendly approach for these carers to work through independently within home/community environment at their own pace and time [18,20].

A pilot controlled trial of a family bibliotherapy for young people with first-episode psychosis in Australia reported that a self-help manual-reading program resulted in marked improvements in first-time carers’ distress, coping skills and positive caregiving experiences over a 16-week follow-up, when compared with usual care [20]. Our research team has translated (into Chinese) and modified the five-module Australian bibliotherapy manual by extending four-month manual-reading to five months and changing telephone follow-ups to group review sessions involving an experienced family carer as facilitator. The modified five-module caregiver-facilitated Problem-solving Based Self-learning Program (PBSP) was initially validated with family care experts (psychiatrists, psychiatric nurses and clinical psychologists) and evaluated in our previous single-site two-arm pilot study (conducted from May 2014 to June 2016) in families of people with recent-onset psychosis in Hong Kong, showing the preliminary effects on the caregiving burden and experiences and patients’ psychotic symptoms immediately and/or six-months post-intervention [9]. The participants’ (carers’) suggestions for improvements (user-friendliness and problem-solving) were integrated into the PBSP to be further tested in the current larger-scale controlled trial for early-stage (≤5 years of onset) psychosis.

This multi-centred randomised controlled trial investigated the effects of a (family) caregiver-facilitated PBSP for Chinese people with early psychosis, when compared with a family psychoeducation group program (FPGP) or usual psychiatric care (UPC) only. It was hypothesised that when compared with FPGP and UPC over six-month follow-up, the PBSP participants could significantly improve in family carers’ perceived burden (primary outcome), caregiving experience and problem-solving, as well as the patients’ subjective recovery, symptom severity, and average number and length of re-hospitalisations.

## 2. Materials and Methods

### 2.1. Study Design

This assessor-blind randomised controlled trial adopted a repeated-measures, three-arm design. Subject recruitment was conducted between March and December 2017, and follow-up was between September 2018 and February 2019. An intention-to-treat principle was used for data analysis, in which all participants were assessed with the same set of outcome measures across three measurements (at recruitment and one-week and six-month post-intervention), regardless of the adherence to their assigned interventions [21]. This was a multi-centred randomised controlled trial (Phase 2) registered at ClinicalTrials.gov (NCT02391649). The original registered study protocol was modified following the initial two-arm pilot study [9] to include the adoption of a smartphone-based manual, a weekly self-record for reading progress and group sessions to discuss difficulties in problem-solving activities. The Consolidated Standards of Reporting Trials (CONSORT) flow diagram [22] of the trial procedure is shown in Figure 1.

### 2.2. Participants and Procedures

The controlled trial was conducted at three Integrated Community Centres for Mental Wellness (ICCMWs) in Hong Kong, which provide community care and support for families of people with mental illness. About 750 families of patients with early-stage psychosis (e.g., brief psychotic disorder, psychotic disorders with featured delusion/hallucination symptoms and delusional disorders; ≤5 years of illness) were attending the three studied ICCMWs (about 250/centre) at recruitment. Of 180 (72%) approached and screened for eligibility in each centre, 80, 85 and 90 (response rates = 44–50%) were found to be eligible and agreed to participate. Thirty-eight participants (48–42%) were randomly selected from their potential subject lists in alphabetical order of surnames, using computer-generated random numbers provided by an independent statistician.

Participants’ inclusion criteria were Chinese adult patients (aged 18–60) who were: attending one of the study centres; diagnosed with psychotic disorders according to the Diagnostic and Statistical Manual, Fourth Text-Revised or Fifth version, DSM-IV-TR [23] or DSM-5 [24]; recent-onset (≤5 years) psychosis at recruitment; and able to understand and communicate in Cantonese/Mandarin. Patients were excluded if they had cognitive impairment and/or learning disability, and/or unstable mental/emotional state and were unfit for study participation as assessed/recommended by a psychiatrist. The family carers were eligible if they were aged 18 or above, had a kinship relationship with and were the main caregiver to the patient at home (>4 h/day and nominated by patient), and spoke in Cantonese/Mandarin.

Upon baseline measurement, participants were randomly assigned to one of the three arms using another set of random numbers generated by the statistician with a stochastic minimisation program (balancing gender, illness duration (<1, 1–3 and >3 years) and symptom severity between groups) [25]. The participant allocation lists were concealed to the researchers, outcome assessors and centre staff.

Sample size was estimated by referring to the effect sizes of perceived caregiving burden between 0.56–0.85 in two similar clinical trials of family self-learning programs in first-episode psychosis [8,20], and two family psychoeducation programs in all-stage psychosis [26,27], in comparison to treatment-as-usual and/or psychoeducation group over the six-month follow-up. As the result, a total of 102 patients (34/group) were required to detect any statistical differences on caregiving burden between three arms at the lowest identified effect size of 0.56, at study power of 0.80 (2-sided, *p* < 0.05). Adding a potential 12% attrition rate based on the average rate of controlled trials of family interventions in psychotic disorders and other mental and physical illnesses [8,26,27,28], the required sample size was 114 (38 per group).

### 2.3. Measures

The primary outcome was caregiving burden (family burden interview schedule, FBIS) [29] and secondary outcomes included carers’ caregiving experience (experience of caregiving inventory, ECI) [30] and problem-solving skills (social problem solving inventory—revised: short version, SPSI-R:S) [31], and patients’ psychotic symptoms (positive and negative syndrome scale, PANSS) [32], progress of recovery (questionnaire about the process of recovery, QPR) [33], and average number and length of re-hospitalisations in the previous five to six months. All of these instruments (FBIS, ECI, PANSS, and QPR) were Chinese versions and/or validated in Chinese patients with schizophrenia or psychotic disorders. The PANSS has shown to have very good test–retest reliability (intra-class correlation = 0.85 to 0.90), internal consistency (Cronbach’s α = 0.88 to 0.91) and concurrent validity with the brief psychiatric rating scale (Pearson’s r = 0.85 to 0.90) in psychotic disorders [14]. The Chinese version of the ECI and its subscales has indicated good test–retest reliability (intra-class correlation coefficients of 0.83 to 0.97) and acceptable to satisfactory internal consistency (Cronbach’s alphas ranged from 0.58 to 0.85) [30]. The Chinese SPSI-R:S has demonstrated a stable five-factor structure (RMSEA = 0.05) with satisfactory internal consistency (Cronbach’s alphas ranged from 0.65 to 0.81) [31]. The Chinese version of the FBIS has also shown adequate test–retest response stability (r = 0.83 for the scale and r = 0.88–0.92 for the domains) and high internal consistency (Cronbach’s alpha of 0.87 for the scale and 0.78–0.88 for its subscales) [34]. Similarly, the Chinese version of the QPR has shown good internal consistency (Cronbach’s alphas were 0.90 for the scale and 0.88 to 0.90 for the subscales) and two-week test–retest reliability (intraclass correlation ranged from 0.87 to 0.92) [33].

Total number and duration of individual patients’ re-hospitalisations and the total number of patients per group being hospitalised over the previous five to six months, and dosages of psychotropic medications, were checked from the clinical records of both the centres and outpatient clinics by a research assistant who was blinded to group assignment.

### 2.4. Interventions

Thirty-eight carers participated in the five-month manual-guided PBSP, in addition to UPC provided by the centres and psychiatric outpatient clinics. They were asked to read and practise caregiving and problem-solving with reference to the five modules of the self-help manual developed by McCann et al. [20,34] in Australia, and its Chinese translated version validated and refined by the research team [9]. The self-help manual guided family carers to learn about the knowledge, community resources, and problem-solving strategies in early-stage psychosis management. The problem-solving approach used is a self-directed cognitive and behavioural process by which the carers identify effective or adaptive ways for resolving important life situations/problems in caregiving [35,36]. The process involved guided learning and resource location by reading the written information and references, followed by exploring and searching for resources by themselves, enabling the carers in this study to solve their problems in caregiving for their relative ‘step-by-step’. The PBSP consisted of five modules (M1–M5; see Table 1), concerning carers’ self-care, seeking services/supports, psychosis care, managing illness impacts on patient and family. The contents of the modules involved reading materials about the illness and its treatments/services, together with case scenarios of caregiving and often-encountered problems. At the end of each module (about 8–10 h for completion), carers were directed to engage in some behavioural rehearsals/exercises on caregiving. Through module reading and exercise completion, carers were supported to develop positive attitudes to caregiving, identifying and consolidating their specific caregiving problems/obstacles, considering and anticipating the implications of alternative problem-solving strategies, and trying out the solution(s) and monitoring it/them as needed.

In addition to module completion over five months by carers themselves, a facilitator (i.e., family carer who was experienced in psychosis care and trained in three full-day sessions by an experienced psychiatric nurse and the researchers based on the manual content) conducted one introductory/engagement group session and three (review) group sessions (1.5 h sessions in the 1st, 6th, 13rd, and 22nd weeks of intervention). During all group sessions, the carers were encouraged to complete one module over the next four weeks; and in the second to fourth group sessions, the facilitator assisted the carers to review their learning progress of module completion, illness management and problem-solving skills in addressing caregiving and related difficulties, with a set of standardised questions. The carers also sought clarifications about the module materials and caregiving issues, and to openly disclose/discuss their family needs and emotions [8]. In addition, the facilitator shared his/her specific caring experiences (both rewarding and challenging), cultural considerations (e.g., stigma towards mental illness), and effective problem-solving strategies in caregiving. The PBSP manual and questions in review sessions were validated by psychiatric rehabilitation experts (psychiatrists, rehabilitation nurses and psychologists) on its clarity (90–98%), relevance and appropriateness (both 93–99%) to family caregiving in psychosis, with minor revisions made on topics of carer’s psychosocial health and interpersonal skills.

Three group sessions of the PBSP and the FPGP were randomly selected and audio-recorded (with carers’ and facilitator’s prior consent) to review and monitor the progress and fidelity (i.e., adherence to the intervention protocols) of the two interventions among the researchers and facilitator(s). Any main problems encountered in the group sessions were discussed and resolved.

The FPGP included twelve 1.5 h sessions held biweekly (similar length/dose of intervention to the PBSP) in group activity rooms of the centres. The program adopted the psychoeducation group manual established by Chan et al.’s [26] and Lehman et al.’s [37] U.S. Schizophrenia Outcome Research Team (PORT) program. The carers received education workshops, caregiving skills training, and psychological support by one advanced practice psychiatric nurse, who had >5 years of experience in psychosis care and rehabilitation. The psychiatric nurse also attended a two-day training workshop (delivered by the researchers) on the essentials of family care for early-stage psychosis, facilitated sharing and discussion in the groups, and supervised practices for group sessions. The FPGP’s four key components/stages of psychoeducation included: (a) program overview and goal setting for psychosis care (two sessions); (b) education and information sharing about psychosis and coping, and survival skills in caregiving (4 sessions); (c) stress management, resilience, and life skills training (4 sessions); and (d) review of learning and related challenges and future plans (two sessions).

UPC consisted of the usual community mental health and psychiatric outpatient services received by all participants, which were similar across centres. Routine community mental health services mainly comprised: (a) medical consultation, treatment and referrals for health and social care by a psychiatrist; (b) brief education on psychosis, medication and treatments/services by a psychiatric nurse; (c) consultation for social welfare and assistance by a medical social worker; and (d) family counselling by clinical psychologist, as needed/referred.

### 2.5. Ethical Considerations

Ethics approval and access for this trial were granted by the Chinese University of Hong Kong (Ref.:CREC2018.319) and the ICCMWs, respectively. The ethics approval was granted on 20 March 2017, at the Research Ethics Committee (KC/KE), Hospital Authority Hong Kong. Informed written consent was obtained from individual participants prior to study enrolment/baseline assessment. All participants were assured of their right to withdraw from the study at any point as well as the confidentiality of personal identity/collected data. All study data and participant lists were locked away in password protected computers with access by the research team only.

### 2.6. Data Collection Process

After obtaining written consent, the research assistant, blinded to group assignment, facilitated the baseline measurement (Time 1) in the centres. Participants were then randomly allocated into one of the three arms by one researcher and asked not to discuss the interventions received with staff and other families in the centres to minimize the risk of treatment contamination [38].

At one-week (Time 2) and six-month (Time 3) post-intervention, the research assistant conducted two post-tests. The information about psychiatric re-hospitalisations over the past five–six months were identified from the centre records. Sociodemographic and clinical data were also collected for both carers and patients, particularly the duration of illness and types and dosages (haloperidol equivalents) [39] of psychotropic drugs taken. The participants and centre staff reported any brief psychological or social therapies provided by the centres or community mental health service during the study.

Intervention fidelity was assessed with a checklist to assess adherence to the contents and instructions of the PBSP and FPGP, as recommended by the National Institute of Health Behaviour Change Consortium [38] by two researchers rating the peer facilitator’s adherence to the manual items with the audio-taped sessions.

### 2.7. Statistical Analyses

All study data were entered, checked, and analysed with SPSS (IBM, Armonk, NY, USA), IBM for Windows, version 23, following the intention-to-treat principle. All characteristics and baseline outcome mean scores of all participants were compared between the three study groups and centres, using one-way ANOVA or the Chi-squared test to examine the homogeneity of groups. There were no significant differences in baseline data between groups/settings and the correlations between baseline outcome scores were low (r = 0.25–0.49). Generalised estimating equations (GEE) examining the changes in each of individual outcomes of the study between the two intervention groups and the control group across the three time-points (Times 1–3) assessed the interaction (Group × Time) treatment effects. The missing data were estimated in the GEE model by using the maximum likelihood estimation without using other replacement methods such as ‘last observation carried forward’ [21]. Pairwise contrast tests were performed to examine any significant mean score differences between groups on each outcome measure at each of the two post-tests if its overall treatment effect was found to be significant in the GEE analysis [40]. Total numbers of patients hospitalised in psychiatric units over the past five–six months, and differences between centres and non-completers (<3 modules and <2 group sessions) versus completers (>3 modules and >2 group sessions) in the PBSP on those significant outcomes were tested using Kruskal–Wallis H and ANOVA tests, respectively. The level of significance was set at *p* < 0.05.

## 3. Results

### 3.1. Characteristics of Study Participants

One hundred and eight of the 114 participants/carers (94.7%) completed their assigned intervention (>3 of 5 self-learning modules and >2 of 4 group sessions), and at least one post-test. Based on the intention-to-treat (ITT) principle, the 106 participants remaining in the study were included in the outcome analyses, excluding those (*n* = 8) who withdrew and did not agree to the use of their data. Two participants in the UPC were unable to be contacted at one-week post-intervention (Time 2) and three to four participants in each of the three study groups declined the six-month follow-up (Time 3). Therefore, there were a total of 21 or 18% of attritions; 13 dropped out and eight withdrew. Figure 1 shows the CONSORT flow diagram of the study procedure. Reasons for withdrawals and/or intervention non-completion mainly included inadequate time for attending/completing the intervention (*n* = 4), patients’ unstable/poor mental state (*n* = 3), and lack of interest/confidence in self-care (*n* = 3). The intervention fidelity for the PBSP and FPGP were very satisfactory (around 92%, ranged 88–94% agreement on item performance), indicating very good adherence to the intervention protocols.

The socio-demographic and clinical characteristics of all participants (family carers and patients) are summarised in Table 2. Carers were on average 34 years old (mean = 32.2–35.8, SD = 8.3–9.0), female, and a parent/spouse of the patient. Patients were mainly male (58–63%), living with one–three family members, and on average were aged 25–26 years (mean= 24.2–26.5, SD = 6.8–7.8). Most of them were taking one oral typical/atypical antipsychotic (71–79%), in low to medium doses (haloperidol equivalents ranged 3.0–8.0 mg/day) [39]. There were no significant differences in all participants’ socio-demographic and clinical characteristics between groups (*p* = 0.11–0.48; see Table 2) and centres (*p* > 0.18). There were also no significant correlations between these characteristics and the baseline outcome scores, and between the baseline outcome scores (Spearman’s r < 0.15). There were also not any significant differences in the mean scores of all baseline outcome scores between three groups (*p* = 0.18–0.30).

### 3.2. Treatment Effects

There were no adverse effects reported for the interventions used. Mean (and SD) values of all outcome scores in all three measurements for the participants who remained in the study (*n* = 106) are presented in Table 3. Only limited missing data (1–2 items of FBIS, QPR and ECI; <2%) were found in two to three participants; without creating noticeable changes in the outcome results, these missing data were imputed by bringing forward the participants’ previous scores/data [40].

The results of GEE test (Table 3) indicated significant interaction (Group × Time) treatment effects on six study outcomes (FBIS, ECI, SPSI-R:S, duration of re-hospitalisations, PANSS, and QPR) between three groups over six-month follow-up, Wald χ^2^ = 7.05–21.87, *p* = 0.02–0.001), with moderate to large effect sizes (η^2^) between 0.12 and 0.24 using repeated-measures ANOVA tests [21]. The group and time effects on the six outcomes in the GEE analysis also indicated similar significant results to the interaction treatment effects as shown in the table.

Pairwise contrast tests following the GEE (Table 4) indicated that, compared with the UPC and/or FPGP, the PBSP reported significant greater improvements in:▪Caregiving burden (FBIS) and experiences (ECI) at Time 2 (mean difference = 4.22 and 19.29, SE = 3.02 and 5.12, *p* = 0.01 and 0.0005, respectively) and Time 3 (MD = 9.12 and 31.97, SE = 3.91 and 8.86, *p* = 0.001 and 0.0001, respectively) than the UPC, and at Time 3 (MD = 3.37 and 9.69, SE = 2.98 and 5.98, both *p* = 0.01, respectively) than the FPGP;▪Problem-solving ability (SPSI-R:S) at Times 2 and 3 (MD = 4.22 and 9.12, SE = 3.02 and 3.91 *p* = 0.01 and 0.001) than the UPC, as well as at Time 3 (MD = 3.37, SE = 2.98, *p* = 0.01) than the PBSP;▪Psychotic symptoms (PANSS) and subjective recovery (QPR) at Time 2 (MD = 26.57 and 6.49, SE = 4.12 and 1.09, *p* = 0.005 and 0.008, respectively) and Time 3 (MD = 50.60 and 13.22, SE = 8.01 and 3.98, *p* = 0.0001 and 0.0003, respectively) than the UPC, and at Time 3 (MD = 15.87 and 5.74, SE = 2.98 and 1.67, *p* = 0.007 and 0.008) than the FPGP;▪Average length or duration (days of hospital-stay) of re-hospitalizations at Time 3 than the UPC (MD = 7.70, SE = 3.97, *p* = 0.007), and FPGP (MD = 5.17, SE = 1.98, *p* = 0.01).

In addition, the participants in the PBSP reported significantly fewer patients being re-hospitalised over the previous five to six months than the other two groups over the six-month follow-up (Kruskal–Wallis test = 6.81, *p* = 0.005; Table 3), in which the PBSP had significantly fewer patients re-hospitalised at Time 3 (*n* = 8, 22%) than both FPGP (*n* = 13, 37%) and UPC (*n* = 20, 57%), i.e., Mann–Whitney U test = 8.4 and 17.9, *p* = 0.05 and 0.001, respectively. However, the above outcome scores, dosages/types of anti-psychotic drugs and other psychosocial interventions received (e.g., family/patient psychotherapy, counselling and medication management) did not show any significant differences between groups (*p* = 0.20–0.38), across centres (*p* = 0.12–0.20), and between the PBSP’s non-completers and completers (*p* = 0.10–0.28), over the follow-up period.

## 4. Discussion

This randomised controlled trial was one of very few problem-solving based self-learning and manual-reading programs for adult family carers of people with recent-onset psychosis. The refined Chinese self-learning manual utilised in the PBSP, together with the facilitation of a trained family carer, can be effective to assist these carers to learn/practice family caregiving for patients with early-stage psychosis in community care. The PBSP has also demonstrated its superiority of treatment effects at six-month follow-up, compared with a well-accepted family psychoeducation group program used in both Western and Chinese communities, in terms of caregiver burden, patients’ symptom severity, and duration of re-hospitalisations. These findings are highly encouraging to support the usefulness of this five-month PBSP for these families in giving care to their relatives with early psychosis, with an overall medium to large effect size at a medium-term (six-month) follow-up. Although the results of the current study demonstrate that the PBSP seems effective in a Chinese setting, further research needs to be conducted in Western or other non-Chinese contexts to establish whether the modifications made to the intervention (i.e., the extended time to complete the manual and family carer facilitator) will result in better outcomes than the standard family psychoeducation group program.

The refined Chinese version of the PBSP showed more positive and diverse treatment effects (on carers’ burden and problem-solving skills and patients’ symptom severity and recovery) than the original English and translated Chinese family bibliotherapy program for carers of people with first-episode psychosis [8,9,15,20]. In comparison with our earlier pilot study [9] the relatively stronger treatment effects observed in the current study might be due to an extended period of manual completion from four to five months, with four additional review (group) sessions facilitated by an experienced family carer to address the participants (carers’) difficulties in caregiving, and/or having a smartphone-based manual for easy access/reading. The presence/use of an experienced peer facilitator in psychosocial interventions, especially in self-learning or community mental health care programs, not only better engaged/empowered participants in group sessions but also overcame the weakness of employing a family therapist who would require extensive training and subsequent resources [12,28,37].

A systematic review and meta-analysis of 12 randomised controlled trials of family intervention (*n* = 1644) in recent-onset psychosis reported that psychosocial interventions such as family psychoeducation and mutual support groups were more effective than usual family support/care in reducing caregiving burden over the long term (up to 36 months), and improving family functioning over short-term follow-up [1,5]. Not similar to the findings of this trial, other important family/patient outcomes of participants (e.g., caregiving skills and experiences, and patient recovery/relapse) in family interventions were not significantly improved. Consistent with two systematic reviews and other family support group studies [5,6,12], the findings of this trial suggest that peer support (family carer facilitator and review group members) might produce better empowerment and motivation to participants (family carers) in problem-solving and caregiving achievements, such as sharing effective coping and caregiving experiences with group members, than the didactic information delivery in psychoeducation and other educational programs in recent-onset psychosis. There was also no increased utilisation of community mental health or family support services found in the PBSP, indicating no additional service demands induced by the PBSP.

Self-help or the self-learning of illness management, together with the use of effective problem-solving strategies in daily family/social situations, are increasingly important and crucial in family interventions in severe mental illnesses [41]. This is particularly essential for those first-time carers of a relative newly diagnosed with mental illness, in order to better equip these carers with new caregiving skills and roles [15,18]. Although the self-learning program in this trial was found effective to improve both family and patient outcomes, the use of a self-help approach to family intervention in psychosis has not yet been adequately studied, especially on its longer-term effects. The therapeutic ingredients and mechanism of the multi-component PBSP not yet examined in this trial should be further investigated.

It is important to note that the family self-learning program has also produced significant benefits to patients with psychosis under care over the six-month follow-up, such as symptom reduction and recovery improvement. These results suggest that the PBSP might enhance the family carers’ motivation and competence in caregiving, resulting in providing a better quality of care and thus improving the patient outcomes. The PBSP has demonstrated a wider variety and more significant levels of improvement in patient and family outcomes than those of family interventions in early-stage psychosis [12,13]. For instance, in addition to significant improvements in caregiving burden and experiences, the patients’ psychotic symptoms in terms of the average PANSS score were reduced from 118.5 at baseline to 28.3 at six-months post-intervention (24.3%), indicating a moderate effect size, Cohen’s *d* = 0.52 [21].

The high completion (about 95% of participants completed at least three of the five self-learning modules and at least two of the four group sessions) and moderately low attrition and withdrawal rates (10.5%) of the PBSP suggests the high acceptability and feasibility of such self-learning programs among family carers. These could be considered much more successful relative to other family interventions in early psychosis (i.e., 50–80% of intervention completion and 12–40% of attritions) [6,11,12]. This self-learning program, which was user-friendly and less costly (in manpower and resources) and lengthy than the conventional psychosocial interventions (e.g., psychoeducation group and cognitive-behavioural therapy) [10,12,41] could be implemented/managed by the carers themselves with only four group sessions, and thus could feasibly be adopted in the current early psychosis service.

Throughout the PBSP, psychiatric nurses adopted a training and supporting role to help the peer facilitator, and participants (family carers) as needed/requested, to achieve their independent self-learning activities. As suggested by the literature [42], nurses play a key role as a resource person to provide professional inputs for the family carers to learn and practise self-management and provide nurse–facilitator–carer collaborative care [5,43] in performing positive and successful caregiving. For the collaborative partnership approach to be widely accepted and adopted, future research can further build on the intervention and findings of this study for the design of an optimal self-learning program/service, and upskilling mental health professionals, to support family carers and their patients during each stage of the collaborative process.

### Limitations

There are a few main limitations of this study. Firstly, the sample (family carers and patients) was recruited from three of 25 ICCMWs in Hong Kong, and the patients’ illness duration was <5 years (and untreated illness period was unavailable). Despite randomisation being performed, the sample was selective/skewed in terms of study venues and sample characteristics (i.e., the participants volunteered to take part, were mainly middle-class families, and patients had early psychosis and less severe symptoms). This might be reflected in the high intervention/study completion rates. Although the sample size was adequate for the current study, further research in this area should be conducted with a larger sample size across multiple settings in order to confirm our findings and establish the efficacy of the intervention in different populations.

Secondly, the participants were not blind to intervention allocation, thus likely producing performance and/or desirable response bias. Although they were asked not to, it was not possible to control whether the participants had discussed their study/intervention participation with centre staff and other families, resulting in potential treatment contamination across groups [22].

Thirdly, although the self-learning progress was reviewed in the four group sessions, there was no opportunity for families to clarify their understanding about the module contents and problem-solving activities outside the group. Therefore, their individualised caregiving needs and concerns might not have been fully addressed. Finally, this was a multi-component intervention, mainly including self-help manual, online resources, peer-facilitator support, and problem-solving practices. The therapeutic effects of individual PBSP components and peer facilitation by a trained family carer were not examined, and thus can be investigated in future research.

## 5. Conclusions

This randomised controlled trial provides evidence that the caregiver-facilitated PBSP in early-stage psychosis can be effective to improve both family carers’ and patients’ psychosocial health, mainly including caregivers’ burden and problem-solving and patients’ symptom severity and recovery. The findings also support the use of the PBSP as a family-oriented intervention during early psychosis in the community-based mental health services in which therapists and resources are limited, and a psychiatric nurse can act as a resource person in the services. Further research on the PBSP is warranted for psychotic patients and their families with diverse characteristics and backgrounds across countries/cultures, and a longer-term follow-up.

## Figures and Tables

**Figure 1 ijerph-17-09343-f001:**
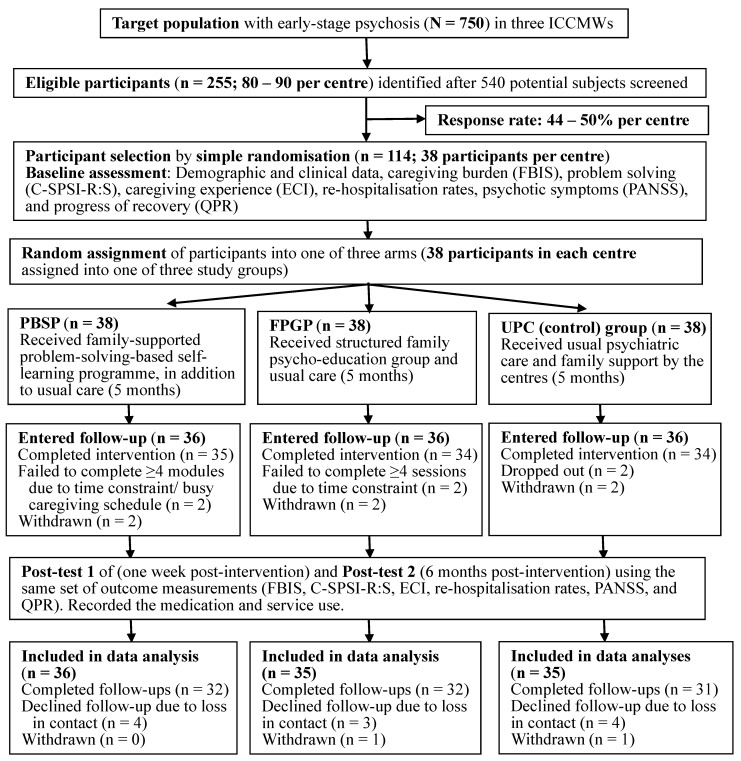
The CONSORT flow diagram of study procedure. Note: ICCMWs: Integrated Community Centres for Mental Wellness; PBSP: problem-solving based self-learning program; FPGP: family psychoeducation group program; UPC: usual psychiatric care only. C-SPSI-R:S: Chinese revised short-form social-problem-solving inventory; ECI: experience of caregiving inventory; FBIS: family burden interview schedule; PANSS: positive and negative syndrome scale; QPR: questionnaire about the process of recovery.

**Table 1 ijerph-17-09343-t001:** Content of problem-solving based self-learning program.

Theme	Goals	Content	Length of Each Theme ^a^
Group session 1			
Orientation and engagement into group	Introduction of problem-solving, self-learning, and the PBSP manual and program, treatment goals and objectives.	Orientation to the program; rationale for using problem-solving based self-help or self-learning, and its manual Establishing trust and goal setting; accepting the roles and responsibilities of a manual user and carer Psychological preparation for psychosis care Resources for important information in the manual, schedule of the self-help program, and telephone support provided by the facilitator, or other professionals, as needed	1 session (1.5 h group session) in the first week
Understanding about the illness and its treatment/care	Understanding psychosis, early stage of the illness and its care; the illness related behaviours, and management	Caregiver’s self-assessment about the understanding and misunderstanding of psychosis Information about the illness and community support resources and the importance of family support, stress coping, and demands in caregiving Identifying the needs for psychosocial care/rehabilitation for patients in the community Discussion about health and caregiving problems encountered and the effects on the patient and family members	
Five reading modules—Improving carer’s well-being and coping skills Introduction: To increase the carer’s understanding of recent-onset psychosis and appreciate how to use the problem-solving framework. Main topics include: What is psychosis? (e.g., symptoms and causes; types and myths; stereotypes and stigma; role of a carer, family and friends; common experiences of caregiving). Introduction to problem-solving approaches and self-assessment; styles and strengths and weaknesses of each psychoeducation approach.			
Module 1: Carer’s well-being	Assisting the carer to work through their emotions, reflect upon how they are currently looking after themselves, and developing good coping strategies in caregiving and family care.	Understanding of most family carers’ emotions and well-being; identify key areas of care burden and stressors encountered by carers Using the four-step ADAPT method of problem-solving in caregiving: **A**—Adopt a positive, optimistic attitude towards the problem **D**—Define correctly the problem by writing out all the facts, identify the obstacles to solving the problem, and specify a realistic goal **A**—(Thinking of) Alternative ways for overcoming the problem and achieving the goal **P**—Predict the consequence that may occur for each alterative **T**—Try-out the problem-solving approach in ‘real-life’ and evaluate whether the chosen approach works	Completing the module over 3 weeks
Group session 2			
Problem-solving activities for carers’ well-being	Assisting carers working through their emotions and challenges in caregiving	Sharing of intense emotions and feelings about mental illness Discussion of ways to deal with negative emotions in group Initial discussion of schizophrenia, health problems and effects to oneself and family	1 session (1.5 h group session) after the first module
Module 2: Benefitting the most from support services	Enable carers to access support services and make the most of these services	Roles and responsibilities of carer (review), access to community support services Confidentiality and seeking financial support Communicating and building relationships with service providers and a framework for asking questions from these providers; asking advice and making complaints/voicing out concerns Understanding approaches to problem-solving and practicing related activities	Completing 1 module over 3–4 weeks
Module 3: Well-being of the person with psychotic disorders	Enhancing carers’ understanding of how to promote their own and patients’ well-being; equipping carers to provide practical and emotional support to patients	Increase in the understanding of a carer’s contribution to treatment and recognising early signs of relapse Promoting well-being of the patient and carer with emotional/practical support Understanding relapse prevention, medication use, stress management, and family support Reinforcing the use of the problem-solving approach	Completing the module over 3 weeks
Group session 3			
Problem-solving for maintaining the well-being of a person with psychosis	Relapse prevention and symptom management	Understanding expressed emotion and negative impacts on patients Effective communication with patients Coping with patients’ hallucinations, delusions, and aggressive behaviours Coping with the negative symptoms (e.g., lack of motivation, social withdrawal) Managing the side-effects of medications	1 session (1.5 h group session) after completing the third module
Module 4: Dealing with the effects of the illness, Part I	Examining effective ways of carer’s communication with patient, equip carer about how to manage a patient who is lacking motivation, socially withdrawn, or engaging in risky/unstrained behaviour	Learning/Practising effective communication skills and motivational interviewing technique Assisting assessment and prevention of social withdrawal, risky behaviours (e.g., aggression and harms), disturbed sleep and psychotic symptoms with problem-solving approach	Completing the module over 3–4 weeks
Module 5: Dealing with the effects of the illness, Part II	Exploring ways for carer to respond to suicidal behaviours, depression and anxiety, suicide, and self-harm	Review carer’s attitudes toward caregiving and conflicts with patient Managing weight gain, reluctance to take medication, substance misuse, depression, and suicidal ideas Learning self-care, relaxation, and alertness of signs for danger or behavioural changes, and seeking help as needed	Completing the module over 3–4 weeks
Group session 4			
Practices, review, and future plans	Behavioural rehearsals of problem solving in caregiving; establishing community resources; self-reflection of learning experiences	Understanding how to best take care of oneself and using learned skills to deal with future problems in thoughts and moods - Identifying signs of relapse and associated factors; reflect on daily activities, stressors and accompanying emotions (i.e., nourishing vs. depleting activities) - Evaluation of self-care, illness management, coping skills and interpersonal relationships - Preparing for future life problems and relapse prevention - Consolidation of selected and practiced coping and mindfulness skills Being familiar with community support services and resources for psychosis care; review of main issues and those skills learned and selected for practices - Summary of the main issues and topics covered, and knowledge and skills learned - Introduction of available community support resources - Anticipated issues in future life and psychological and behavioural preparations Self-reflection of learning experiences - Review of self-care and coping skills learned and practiced in family situations - Summing up of knowledge, attitude, and practices in illness management - Making plans for future life and psychosis care	1 session (1.5 h group session) after completing the fifth module

PBSP, problem-solving based self-learning program. **ADAPT** is bolded and denoted Adapt, Define, Alternative, Predict, and Try-out. ^a^ The PBSP comprised 5 modules of a self-help manual intertwined with 4 face-to-face group review sessions and 4 telephone follow-ups over 5 months.

**Table 2 ijerph-17-09343-t002:** Characteristics of PBSP, FPGP and UPC participants (*n* = 114).

Characteristics	PBSP (*n* = 38) f (%) or M ± SD	FPGP (*n* = 38) f (%) or M ± SD	UPC (*n* = 38) f (%) or M ± SD	Chi-Squared or ANOVA Test ^#^	*p*
Family carers					
Gender				χ^2^ = 1.02	0.48
Male	14 (36.8)	12 (31.6)	12 (31.6)		
Female	24 (63.2)	26 (68.4)	26 (68.4)		
Age (years)	32.2 ± 8.8 CI = 26.1–39.3	33.9 ± 8.3 CI = 27.6–42.6	35.8 ± 9.0 CI = 25.8–44.9	F = 2.18	0.11
18–25	14 (36.8)	12 (31.6)	13 (34.2)		
26–30	14 (36.8)	15 (39.5)	14 (36.8)		
31–35	5 (13.2)	6 (15.8)	6 (15.8)		
>35	4 (10.5)	5 (13.2)	5 (13.2)		
Education level				χ^2^ = 1.89	0.23
Primary school or below	9 (23.7)	8 (21.1)	10 (26.3)		
Secondary school	20 (52.6)	19 (50.0)	20 (52.6)		
University or above	9 (23.7)	11 (28.9)	8 (21.1)		
Relationship with patient				χ^2^ = 1.28	0.30
Child	8 (21.0)	7 (18.4)	7 (18.4)		
Parent	12 (31.6)	12 (31.6)	14 (36.9)		
Spouse	13 (34.2)	14 (36.8)	13 (34.2)		
Others (e.g., sibling)	5 (13.2)	5 (13.2)	4 (10.5)		
Monthly household income (HKD) ^~^	16,481 ± 4123 CI = 11,323–21,205	17,912 ± 513 CI = 11,211–23,533	17,367 ± 4902 CI = 12,250–22,887	F = 1.81	0.14
10,000 or below	7 (18.4)	5 (13.2)	6 (15.8)		
10,001–20,000	17 (44.7)	15 (39.5)	16 (42.1)		
20,001–30,000	9 (23.7)	11 (28.9)	9 (23.7)		
>30,000	5 (13.2)	7 (18.4)	7 (18.4)		
Patients					
Gender				χ^2^ = 1.64	0.16
Male	22 (57.9)	23 (60.5)	24 (63.2)		
Female	16 (42.1)	15 (39.5)	14 (36.8)		
Age (years)	24.2 ± 6.8 CI = 18.0–30.3	26.2 ± 7.8 CI = 18.8–35.3	26.5 ± 7.7 CI = 28.5–36.5	F = 1.25	0.24
18–24	17 (44.7)	16 (42.1)	15 (39.5)		
25–30	15 (39.5)	17 (44.7)	18 (47.4)		
31–37	6 (15.8)	5 (13.2)	5 (13.2)		
Education level				χ^2^ = 1.65	0.20
Primary school or below	8 (21.1)	7 (18.4)	7 (18.4)		
Secondary school	20 (52.6)	21 (55.3)	19 (50.0)		
University or above	10 (26.3)	10 (26.3)	12 (31.6)		
Duration of illness (months)	10.1 ± 4.8, range = 4.0–19.5 CI = 4.5–15.2	11.9 ± 5.9, range = 3.5–19.0 CI = 7.2–18.7	12.8 ± 6.7, range = 4.0–21.5 CI = 4.1–21.0	F = 1.70	0.23
3–6	8 (21.1)	9 (23.7)	7 (18.4)		
7–12	16 (42.1)	15 (39.5)	17 (44.7)		
13–18	10 (26.3)	11 (28.9)	11 (28.9)		
19–22	4 (10.5)	3 (7.9)	3 (7.9)		
Number of family members living with patient				χ^2^ = 1.42	0.28
One	12 (31.6)	14 (36.8)	15 (39.5)		
2–3	20 (52.6)	20 (52.6)	18 (47.4)		
4–5	6 (15.8)	4 (10.5)	5 (13.1)		
Types of psychotropic drugs				χ^2^ = 1.86	0.21
Conventional antipsychotics (e.g., haloperidol)	14 (36.8)	12 (31.6)	13 (34.2)		
Atypical antipsychotics (e.g., risperidone)	16 (42.1)	15 (39.5)	15 (39.5)		
Anti-depressants (e.g., fluoxetine)	5 (13.1)	6 (15.8)	4 (10.5)		
Blended mode ^^^	3 (7.9)	5 (13.1)	6 (15.8)		
Use of psychiatric services				χ^2^ = 1.52	0.28
Medical consultation and treatment planning	36 (94.7)	38 (100.0)	36 (94.7)		
Nursing advice on services and brief education	38 (100.0)	36 (94.7)	35 (92.1)		
Social welfare and financial advices	25 (65.8)	20 (52.6)	17 (44.7)		
Individual/family counselling	18 (47.4)	20 (52.6)	10 (26.2)		
Dosage of medication ^+^				χ^2^ = 1.32	0.26
High	8 (21.1)	7 (18.4)	6 (15.8)		
Medium	15 (39.5)	17 (44.7)	18 (47.4)		
Low	15 (39.5)	14 (36.9)	14 (36.8)		

Note: PBSP, problem-solving based self-learning program; FPGP, family psychoeducation group program; UPC, usual psychiatric outpatient care. CI, 95% confidence interval. ^#^ An analysis of variance test (F-test, df = 112) was used to compare the participants’ socio-demographic variables in interval/continuous level of measurement between the 3 groups; otherwise, Chi-squared tests were used. ^~^ USD 1 = HKD 7.80. ^^^ Patients were taking >1 type of psychotropic medication such as either conventional and atypical antipsychotics or atypical antipsychotics and anti-depressant. ^+^ Dosages of psychotropic/antipsychotic medications compared with the average dosage of medication taken by patient in haloperidol-equivalent mean value; High: >8 mg/day, Medium: 4–8 mg/day, Low: <4 mg/day [39].

**Table 3 ijerph-17-09343-t003:** Outcome measure scores at pre-test and two post-tests and results of repeated-measures ANOVA test (*n* = 106).

	PBSP (*n* = 36)			FPGP (*n* = 35)		UPC (*n* = 35)		GEE Analysis		
	Time	M ± SD	95% CI	M ± SD	95% CI	M ± SD	95% CI	Group Effect	Time Effect	Group × Time Effect
Instrument								*β* (95% CI), *p*	*β* (95% CI), *p*	*β* (95% CI), *p*, Wald χ^2^, ES
FBIS (0–50) ^$^	Time 1	29.20 ± 6.42	22.61, 35.83	29.92 ± 5.01	23.0, 38.0	29.82 ± 6.52	22.10, 37.04	0.62 (0.32, 0.95), 0.005	−0.68 (−0.98, −0.38), 0.008	−2.02 (−4.48, −0.12), 0.001 Wald χ^2^ = 19.43, ES = 0.20
	Time 2	25.41 ± 5.81	19.52, 31.14	27.04 ± 5.95	19.9, 31.5	29.63 ± 7.12	22.01, 38.05			
	Time 3	22.82 ± 5.02	17.02, 29.12	26.13 ± 7.12	19.2, 33.5	31.94 ± 8.01	23.63, 40.05			
ECI (0–204) ^#^	Time 1	130.21 ± 16.43	114.1, 146.92	133.22 ± 16.52	116.52, 149.83	133.02 ± 18.42	114.00, 150.54	0.42 (0.21, 0.63), 0.01	−0.60 (−0.98, −0.22),0.01	−1.41 (−2.03, −0.89), 0.005 Wald χ^2^ = 14.12, ES = 0.18
	Time 2	118.23 ± 21.04	98.23, 140.82	119.83 ± 20.04	98.82, 141.42	137.52 ± 22.02	115.20, 158.54			
	Time 3	109.84 ± 16.41	88.22, 118.43	119.53 ± 18.81	99.23, 138.05	141.81 ± 19.21	120.20, 160.42			
SPSI-R:S (0–100)	Time 1	51.22 ± 8.46	43.12, 58.93	50.23 ± 7.03	41.11, 58.24	50.02 ± 8.04	41.78, 58.26	0.36 (0.18, 0.54), 0.05	0.40 (0.21, 0.59), 0.05	1.01 (0.78, 1.24), 0.05 Wald χ^2^ = 7.05, ES = 0.12
	Time 2	56.23 ± 9.02	47.24, 66.05	53.02 ± 9.16	43.80, 63.06	48.01 ± 8.51	40.20, 57.02			
	Time 3	54.85 ± 8.43	45.63, 63.44	50.82 ± 9.05	42.20, 60.24	47.83 ± 9.23	28.20, 47.26			
Re-hospitalisations										
- Average Number ^~^	Time 1	1.72 ± 0.93	0.62, 2.74	1.62 ± 1.01	0.50, 2.61	1.60 ± 0.90	0.60, 2.70	0.21 (0.10, 0.32), 0.09	0.18 (0.10, 0.26), 0.14	0.74 (0.49, 0.99), 0.10 Wald χ^2^ = 2.05, ES = 0.02
	Time 2	1.43 ± 0.81	0.63, 2.22	1.53 ± 1.00	0.51, 2.62	1.82 ± 1.14	0.68, 3.06			
	Time 3	1.34 ± 0.86	0.54, 2.12	1.72 ± 1.22	0.52, 3.22	1.92 ± 1.31	0.61, 3.23			
- Duration ^^^	Time 1	16.52 ± 5.85	10.52, 22.51	17.01 ± 6.44	11.12, 23.84	15.91 ± 7.12	7.79, 24.03	−0.58 (−0.78, −0.18), 0.01	−0.49 (−0.74, −0.25), 0.04	−1.05 (−1.65, −0.45), 0.02 Wald χ^2^ = 7.08, ES = 0.13
	Time 2	14.02 ± 6.21	6.92, 20.82	15.21 ± 9.02	6.01, 24.25	17.83 ± 9.51	8.52, 27.24			
	Time 3	11.85 ± 8.53	3.52, 21.04	17.02 ± 9.04	7.78, 26.82	19.55 ± 10.81	8.54, 30.06			
No. of patients being hospitalised	Time 1	(17) ^+^		(18)		(17)				KW = 6.81 ^@^, 0.005
	Time 2	(11)		(14)		(16)				
	Time 3	(8)		(13)		(20)				
PANSS (43–215)	Time 1	116.53 ± 17.82	98.52, 135.53	107.22 ± 14.71	96.51, 123.23	118.12 ± 9.81	109.01, 128.23	−0.68 (−0.98, −0.38), 0.005	−0.72 (−1.13, −0.29), 0.002	−1.36 (−1.98, −0.73), 0.001 Wald χ^2^ = 21.87, ES = 0.24
	Time 2	99.64 ± 19.24	80.03, 119.02	99.81 ± 12.21	86.00, 112.02	126.21 ± 17.10	107.99, 142.21			
	Time 3	88.22 ± 17.05	71.03, 106.25	104.11 ± 19.51	84.20, 124.23	138.82 ± 19.81	118.81, 159.03			
QPR (0–88)	Time 1	38.92 ± 9.01	28.83, 38.02	39.21 ± 9.10	30.01, 48.31	38.12 ± 8.50	29.58, 37.02	0.60 (0.23, 0.87), 0.006	0.58 (0.25, 0.91), 0.01	1.64 (1.26, 1.98), 0.005 Wald χ^2^ = 15.28, ES = 0.18
	Time 2	43.57 ± 9.82	32.42, 43.84	40.81 ± 8.21	32.20, 49.42	37.08 ± 9.81	26.07, 46.89			
	Time 3	48.24 ± 11.05	37.02, 59.53	42.50 ± 9.22	32.18, 51.73	35.02 ± 8.31	26.01, 43.63			

Note: PBSP, problem-solving based self-learning program; FPGP, family psychoeducation group program; UPC, usual psychiatric outpatient care; ECI, experience of caregiving inventory; FBIS, family burden interview schedule; SPSI-R:S, social problem solving inventory—revised: short version; PANSS, positive and negative syndrome scale; QPR, questionnaire about the process of recovery. Time 1, baseline measurement at the start of intervention; Time 2, one-week post-intervention; Time 3, 6-month post-intervention. ES, effect size in terms of eta squared (η2) using repeated-measures ANOVA test; whereas η^2^ < 0.06 is a small effect, 0.06–0.13 is a moderate effect and 0.14 or higher is large effect. ^$^ Possible range of scores of each scale in parentheses. ^#^ For ECI, the higher the mean score, the more negative the appraisal of family carers to their caregiving experiences. ^~^ Average number of readmissions to a psychiatric hospital or inpatient unit over the past 6 months at the three measurements (Times 1 to 3). ^^^ Duration/length of readmissions to a psychiatric ward/unit in terms of average number of days of hospital stay over the past 5–6 months at Times 1 to 3. ^+^ Total number of patients per group being hospitalised over the past 6 months at Times 1–3 indicated in parentheses. ^@^Value of Kruskal–Wallis test was used to compare the total number of patients hospitalised across Times 1–3.

**Table 4 ijerph-17-09343-t004:** Post-hoc comparisons of study outcomes betweeb groups at two post-tests.

**Contrast Tests of Mean Scores of Study Outcomes between Groups by Post-Tests**								
**Outcomes**	**Groups**	**Comparison**	**Mean Difference**	**Standard Error**	**95% Confidence Intervel**		**Slope**	***p*** **-Value**
					**Upper**	**Lower**		
FBIS	PBSP vs. UPC	T2	−4.22	3.02	−7.22	−1.81	−4.89	0.01
		T3	−9.12	3.91	−13.05	−5.88	−8.12	0.001
	PBSP vs. FPGP	T2	−1.63	2.21	−3.84	0.58	1.01	0.08
		T3	−3.37	2.98	−6.35	−0.39	−3.07	0.01
ECI	PBSP vs. UPC	T2	−19.29	5.12	−2.41	−14.16	−10.13	0.0005
		T3	−31.97	8.86	−40.83	−23.11	−18.12	0.0001
	PBSP vs. FPGP	T2	−1.60	3.01	−4.61	1.41	−0.13	0.12
		T3	−9.69	5.98	−14.67	−3.71	−4.15	0.01
SPSI-R:S	PBSP vs. UPC	T2	8.22	3.45	11.66	4.77	7.01	0.005
		T3	7.02	2.02	9.05	4.98	5.98	0.008
	PBSP vs. FPGP	T2	3.21	0.87	4.10	2.35	2.71	0.07
		T3	4.03	1.02	5.06	3.00	3.58	0.04
**Post-Hoc Comparisons of Study Outcomes (Continued)**								
**Outcomes**	**Groups**	**Comparison**	**Mean Difference**	**Standard Error**	**95% Confidence Intervel**		**Slope**	***p*** **-Value**
					**Upper**	**Lower**		
Duration of Re-hospitalisations ^^^	PBSP vs. UPC	T2	−3.81	2.90	−6.71	−0.88	−1.01	0.06
		T3	−7.70	3.97	−11.66	−3.74	−6.08	0.007
	PBSP vs. FPGP	T2	−1.19	1.09	−2.20	−0.10	−0.91	0.07
		T3	−5.17	1.98	−7.16	−3.18	−4.01	0.01
PANSS	PBSP vs. UPC	T2	−26.57	4.12	−30.70	−22.45	−9.89	0.0005
		T3	−50.60	8.01	−58.60	−42.48	−13.43	0.0001
	PBSP vs. FPGP	T2	−0.17	1.02	−1.20	0.85	−0.90	0.19
		T3	−15.89	2.98	−18.88	−12.93	−5.76	0.007
QPR	PBSP vs. UPC	T2	6.49	1.09	7.59	5.39	6.01	0.008
		T3	13.22	3.98	16.21	9.30	10.89	0.0003
	PBSP vs. FPGP	T2	2.76	0.91	3.68	1.83	3.12	0.06
		T3	5.74	1.67	7.42	4.08	5.70	0.008

Note: PBSP, problem-solving based self-learning program; FPGP, family psychoeducation group program; UPC, usual psychiatric outpatient care; ECI, experience of caregiving inventory; FBIS, family burden interview schedule; SPSI-R:S, social problem solving inventory—revised: short version; PANSS, positive and negative syndrome scale; QPR, questionnaire about the process of recovery. Time 1, baseline measurement at the start of intervention; Time 2, one-week post-intervention; Time 3, 6-month post-intervention. ^^^ Duration/length of readmissions to a psychiatric ward/unit in terms of average number of days of hospital stay over the past 5–6 months at Times 1 to 3.

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
