# Peer review of "A Randomised Controlled Trial of a Caregiver-Facilitated Problem-Solving Based Self-Learning Program for Family Carers of People with Early Psychosis"

_ijerph, 2020, doi:10.3390/ijerph17249343_

Round 1
Reviewer 1 Report
This is a potentially interesting study on family intervention in psychosis in early stages, and its beneficial aspects for the patient and the family on multiple levels: emotional, social, burden, conflict resolution, coping skills, stigma, etc.
The article has a good approach and uses a validated tool that reinforces the results found and the conclusions they confirm. It has a correct statistical treatment of the data and these are reliable.
To improve the quality, the following issues need to be addressed:
- The abbreviation should be taught once in the text, and from there, always use it, such as the Problem-solving based Self learning Program (PBSP).
- Line 37. "Psychosis is a major disabling and disruptive mental illness accounting for over 30% of psychiatric patient populations in Hong Kong and globally". That statement should be referenced.
- Line 77-78 "by extending 4-month manual-reading to five months and changing telephone". Numbers and words are used interchangeably to refer to numerical units. The criteria must be unified throughout the text.
- Line 157-158 "psychotic disorders with very satisfactory content validity, internal consistency and test-retest and inter-rater reliability [14, 29, 30, 33]". It would be interesting to specify the psychometric properties of the Chinese versions.
- Line 311 "using repeated-measures ANOVE test (Stevens, 2009)". The citation format must be corrected.
- Table 4 there is a mistake with the bold type of "FBIS t2". Also, the column "lower" is apparently not visually balanced.
- The limitations should have a greater impact on the need to increase the sample in future studies.
- Although they are correct and, therefore, it is not necessary to modify, simply a tip that you can value: There are three dense sections that, without losing the essence, I would look for a way to synthesize to facilitate and speed up reading: 2.2. Participants and Procedures. 2.4. Interventions. And 3.1. Characteristics of study participants.
Overall, I found this study to be interesting.
Author Response
Responses to reviewers
Thank you for your valuable and constructive comments and feedbacks. We have responded to each of the comments and indicated changes made onto the manuscript in the table of the attached file. We have also highlighted text changes in the manuscript using red colored fonts.
We hope the manuscript is now acceptable for publication. Thank you.
With best wishes,
WT Chien (corresponding authors)

Reviewer 2 Report
Abstract and Introduction:
The introduction of the paper has been well-written. It summarizes the current research in the area as well as makes a good case for the present study.
Methodology:
Methodology mentions all the relevant information and is easy to understand. 1) It is unclear why authors chose to use DSM-IV TR criteria for sample selection even though DSM-V was available at the time data for this study was collected. Please provide relevant explanation for that choice.
Results:
The results section of the paper is good and has a nice flow; however, few suggestions could be incorporated. 2) Were any significant correlations between demographic characteristics of the study participants (education level, etc.) and the dependent variables being studied?
Discussion:
Discussion has been well-written and previous research has been integrated along with explaining the findings of present study.
Author Response
The response to the second reviewer is attached.

Reviewer 3 Report
The study evaluates the efficacy of a self-help intervention for caregivers of people with psychosis in the first 5 years after diagnosis. The subject is extremely relevant and the results have an almost direct transfer to society.
The article is superbly written; despite the complexity of the design, the authors manage to explain with great clarity all the methodological and content aspects. The analyzes carried out are very adequate, and the discussion includes the main results and limitations of the study.
Personally, I would have been interested in the process of cultural adaptation of the intervention to the Chinese population. The changes in the program in terms of structure,compared to the first Australian version are described very well, but ... to what extent could this program be applied in other countries or cultures? I have missed some comment on this question.
In short, a magnificent job, really interesting and with very relevant results for caregivers of people with psychosis, who are in such need of support and training.
Author Response
The responses to the third reviewer's comments are attached.

Reviewer 4 Report
In general this is a well written paper with good study design, methodology, presentation of results and discussion.
The topic of the paper, support family carers to cope with caregiving for people with severe mental illness, is very interesting and important.
In only have one minor comment:
Why the Authors used a method of haloperidol equivalents of dosages of psychotropic drugs taken? And not perhaps the DDD Method, as proposed here:
Leucht S, Samara M, Heres S, Davis JM (2016) Dose Equivalents for Antipsychotic Drugs: The DDD Method. Schizophrenia bulletin 42 (suppl_1):S90-S94. doi:10.1093/schbul/sbv167
Author Response
The responses to the fourth reviewer's comments are attached.

This manuscript is a resubmission of an earlier submission. The following is a list of the peer review reports and author responses from that submission.